# A Study of the Synthesis and Characterization of New Acrylamide Derivatives for Use as Corrosion Inhibitors in Nitric Acid Solutions of Copper

**DOI:** 10.3390/nano12203685

**Published:** 2022-10-20

**Authors:** Ahmed Abu-Rayyan, Badreah Ali Al Jahdaly, Huda S. AlSalem, Nahlah A. Alhadhrami, Amira K. Hajri, Abeer Abdulaziz H. Bukhari, Mohamed M. Waly, Aya M. Salem

**Affiliations:** 1Chemistry Department, Faculty of Arts & Science, Applied Science Private University, P.O. Box 166, Amman 11931, Jordan; 2Chemistry Department, Faculty of Applied Science, Umm Al-Qura University, P.O. Box 24230, Makkah 21955, Saudi Arabia; 3Department of Chemistry, College of Science, Princess Nourah Bint Abdulrahman University, P.O. Box 84428, Riyadh 11671, Saudi Arabia; 4Chemistry Department, Faculty of Science, Taibah University, P.O. Box 30002, Medina 42353, Saudi Arabia; 5Department of Chemistry, University College Alwajh, University of Tabuk, Tabuk 71421, Saudi Arabia; 6Department of Chemistry, Faculty of Science, University of Tabuk, Tabuk 71421, Saudi Arabia; 7Department of Chemistry, Faculty of Science, New Mansoura University, Mansoura 35516, Egypt; 8Department of Basic Science, Higher Institute of Electronic Engineering (HIEE), Belbis 11621, Egypt

**Keywords:** corrosion inhibition, copper, nitric acid, acrylamide derivatives, langmuir isotherm

## Abstract

The objective of this research was to explore the impact of corrosion inhibition of some synthetic acrylamide derivatives 2-cyano-*N*-(4-hydroxyphenyl)-3-(4-methoxyphenyl)acrylamide (ACR-2) and 2-cyano-*N*-(4-hydroxyphenyl)-3-phenylacrylamide (ACR-3) on copper in 1.0 M nitric acid solution using chemical and electrochemical methods, including mass loss as a chemical method and electrochemical impedance spectroscopy (EIS) and potentiodynamic polarization (PP) as electrochemical methods. By Fourier-transform infrared spectroscopy (FTIR), proton nuclear magnetic resonance (1HNMR), and mass spectroscopy (MS) methods, the two compounds were verified and characterized. There is evidence that both compounds were effective corrosion inhibitors for copper in 1.0 M nitric acid (HNO_3_) solutions, as indicated by the PP curves, which show that these compounds may be considered mixed-type inhibitors. With the two compounds added, the value of the double-layer capacitance was reduced. In the case of 20 × 10^−5^ M, they reached maximum efficiencies of 84.5% and 86.1%, respectively. Having studied its behavior during adsorption on copper, it was concluded that it follows chemical adsorption and Langmuir isotherm. The theoretical computations and the experimental findings were compared using density functional theory (DFT) and Monte Carlo simulations (MC).

## 1. Introduction

The problem of corrosion annually causes losses from closing and stopping facilities, the deterioration of several industries, and accidents due to the collapse of bridges and dams [1,2,3]. Due to the different uses of copper and its alloys in industries, etc., it was necessary to protect the copper surface against the occurrence of corrosion. Although copper is a noble element, it is subject to erosion by strong acids, and alkaline solutions, where copper is dissolved, and a layer of oxide is formed on its surface [4]. 

Inhibitors are arguably the most practical corrosion methods of protection. Inhibitors may be organic materials with some functional groups such as N, S, or O or heterogeneous rings [5,6,7]. These compounds absorb onto the surface and reduce corrosion activity [8,9]. In addition, the N, S, and O atoms may form chemical bonds with the surface and contribute to protecting the metal surface [10,11]. Therefore, using inhibitors in metal protection is the first line of defense against corrosion in the oil and gas industries [12,13]. Copper is one of the most frequent elements in the Earth’s crust. It is an extensively used substance in various industries due to its copper corrosion resistance properties and high conductivity of heat and electricity. Therefore, it is commonly used in producing wires, plates, and pipelines. The corroded surface of copper affects its performance in various industries.

Using natural polymers as corrosion inhibitors is profitable, environmentally friendly, and non-harmful. Inhibitors of this type can also be synthesized easily, including grafted and modified polysaccharides. Polymeric compounds are better at adsorbing than monomers since polymers have multiple adsorption sites [14]. It is possible to use various strategies in the field of polymer coating, ranging from environmentally friendly vegetable oil-based coatings to epoxy acrylates of vegetable origin, as well as biodegradable lignin to engineer the bio–nano interface using multifunctional coordinating polymer coatings. Using ultraviolet light to initiate photochemical reactions is a faster and greener alternative [15]. Additionally, magnetic nanogels are nanosized magnetic polymer composites with a core–shell structure, consisting of nanosized magnetic nanoparticles encapsulated in polymer gel shells, displaying the controllability of magnetic nanoparticles and the swelling ability of hydrogels. Through the crosslinking of the polymer solution, magnetic nanoparticles can be embedded into the hydrogel to produce magnetic nanogels. In this way, the shell protects the metal oxide core from oxidation and aggregate formation [16,17,18].

Various studies yielded a good number of organic compounds used as corrosion inhibitors owing to the existence of N, S, and O atoms and double bonds in them. Heterocyclic compounds are the most important of these compounds. Recently, heterogeneous organic compounds containing N, S, and O atoms have been of interest to researchers to find copper corrosion inhibitors in various acid solutions. These compounds are of great interest due to their adsorption properties and the creation of an oxide layer on the metal surface [19,20]. Density functional theory explains how the metal surface is protected by the adsorption of this substance on the surface by the electron cloud of the π-electron density from the delocalized region through its highest occupied molecular orbital (HOMO) to the metal’s lowest unoccupied molecular orbital (LUMO). Monte Carlo simulations were applied to properly understand the inhibitor’s adsorption behavior on the metal surface.

This study aims to focus on the evaluation of the potency of eco-friendly, nontoxic novelty synthetic compounds, namely a 2-cyano-*N*-(4-hydroxyphenyl)-3-(4-methoxyphenyl)acrylamide (ACR-2) and 2-cyano-*N*-(4-hydroxyphenyl)-3-phenylacrylamide (ACR-3) against the copper dissolution in an acidic medium (1.0 M nitric acid solution) by using chemical and electrochemical methods as well as characterization techniques from Monte Carlo simulations and density functional theory (DFT) to discover the adsorption type and corrosion mechanism on the metal surface.

## 2. Materials and Methods

The experimental calculations were conducted in 1.0 M HNO_3_ solution which was purchased from the El-gomhouri Company in Egypt, in the absence and presence of various concentrations of ACR-2 and ACR-3. Table 1 contains the chemical composition of the copper alloy used in the experiment.

### 2.1. Inhibitors Preparation

#### 2.1.1. Synthesis of 2-Cyano-*N*-(4-hydroxyphenyl)-3-(4-methoxyphenyl)acrylamide

One drop of piperidine was added to acetamide 1 (0.35 g, 2 mmol) and 4-methoxy benzaldehyde (0.22 g, 2 mmol) in ethanol, followed by 1 hour of reflux. Using ethanol and DMF (4:1) as a mixture, the filtered solid product was crystallized.

#### 2.1.2. Synthesis of 2-Cyano-*N*-(4-hydroxyphenyl)-3-phenylacrylamide

Benzaldehyde (0.21 g, 2 mmol) and acetamide 1 (0.35 g, 2 mmol) were dissolved in ethanol, and the reaction mixture was refluxed for 1 hour in the presence of catalytic piperidine. After filtration, EtOH was added for washing, and the solid product was crystallized by ethanol and DMF mixture (5:1) (Figure 1).

### 2.2. Mass Loss Tests (ML)

For the measurements of weight loss, copper samples with a thickness of 2.0 × 2.0 × 0.2 cm were used, which were polished to different degrees of sandpaper, then washed with alcohol to remove impurities and dirt, distilled water, and then the process of good drying of the samples, followed by weight loss measurements by the usual methods at different temperatures which calculated according to following Equation (1) [21,22]: (1)$$\%\eta =\left[1-\frac{W}{W^{{}^{\circ}}}\right]\times 100=\theta \times 100$$

*W* and *W*° are the mass loss of metal without and with investigated compounds, respectively.

### 2.3. Electrochemical Tests

Electrochemical calculations were made at a temperature of 25 °C and using a three-electrode system of platinum, calomel, and copper. In this system, the platinum electrode was employed as a counter, and the calomel electrode as a reference. In the end, the copper electrode was used as the working electrode [23]. To obtain a semi-stable state, the copper electrode was dipped at an open circuit voltage in the presence of an acidic medium. In this case, impedance was measured at different times, with varying frequencies from 10^−2^ to 10^−5^ Hz in the presence of alternating current signals of 5 mv; then, the inhibition efficiency was measured by the following equation [24].
(2)$$\eta \%=\frac{R_{ct}-R_{ct,\;0}}{R_{ct}}\times 100$$

*R_ct_* and *R_ct_*_,0_ are charge-transfer resistances of inhibited and uninhibited copper, respectively. 

For measurements of the polarization value, experiments were carried out at 1 mV s^−1^ and within the range of −250 mV. Inhibitory efficiency values were calculated through the following equation [25]:(3)$$\eta \%=\frac{i_{corr,0}-i_{corr}}{i_{corr,0}}\times 100$$

*i_corr_*_,0_, and *i_corr_* mean current density values lack the existence of the protection of compounds. With a computer monitor attached to Potentiostat/Galvanostat/ZRA. A Gamry framework system based on the ESA 400 is included. Gamry applications also include DC105 polarization software and the EIS300 EIS software, Gamry Instruments, PA, USA.

### 2.4. Theoretical Calculations

The DFT was computed using Gaussian 09W software, Gaussian, Inc., CT, USA The B3LYP approach optimized the acrylamide derivatives on a 6-311++G basis set (d, p). Based on highly and lightly occupied and unoccupied orbitals (HOMO and LUMO), detailed analyses were carried out on the parameters of quantum chemicals (HOMO and LUMO) [26,27]. During the molecular dynamic (MD) simulation, a Forcite unit from Accelrys Inc. (Vélizy-Villacoublay, FR, France) in England was used. The adsorbed molecules were simulated and determined using a copper box. The box included 250 fractions of water and one fraction of the inhibitor. According to their binding and interaction energies, copper and inhibitor can be described as follows:(4)$$E_{interact}=E_{tot}-\left(E_{sub}+E_{Acrylamide}\right)$$
(5)$$E_{Binding}=-E_{interact}$$

*E_subs_* is the total energy of Cu (1 1 1) and H_2_O particles, *E_Acrylamide_* signifies free inhibitor energy, and *E_tot_* embodies the total energy of the total system.

### 2.5. SEM

SEM was used to examine the surfaces of the original Cu sample that was mechanically polished and the Cu sample that was naturally submerged for five hours in a solution containing an inhibitor [28]. Test with 1.0 M HNO_3_, 1.0 M HNO_3_ + 20 × 10^−5^ M of the tested compounds.

## 3. Results

### 3.1. 2-Cyano-N-(4-hydroxyphenyl)-3-(4-methoxyphenyl)acrylamide Characteristics

#### 3.1.1. FTIR of Compound (ACR-2)

The IR of compound (ACR-2) displayed absorption bands at 3406, 3353, 2217, and 1675 cm^−1^, corresponding to hydroxyl, amino, cyano, and carbonyl groups, respectivelyas in Figure 2.

#### 3.1.2. Proton Nuclear Magnetic Resonance (HNMR) of Compound (ACR-2)

The ^1^H NMR showed a singlet signal at *^TM^* 3.87 ppm, corresponding to the methoxy group, in addition to multiplet signals at *^TM^* 6.75–8.02 ppm for aromatic protons. Moreover, singlet signals were found at *^TM^* 8.16, 9.34, and 10.03 ppm for the methine, OH, and NH protons, respectively as in Figure 3.

#### 3.1.3. Mass Spectroscopy of Compound (ACR-2)

MS *m*/*z* (%): 295 (M++1, 23.90), 294 (M+, 47.97), 279 (36.83), 242 (49.26), 202 (54.60), 197 (100.00), 159 (87.57), 145 (63.55), and 88 (57.60). The MS exhibited a molecular ion peak at *m*/*z* = 294 (M^+^, 47.97%) congruous to a molecular formula C_17_H_14_N_2_O_3_ as in Figure 4.

### 3.2. 2-Cyano-N-(4-hydroxyphenyl)-3-phenylacrylamide Characteristics

#### 3.2.1. FTIR of Compound (ACR-3)

The IR of the compound (ACR-3) showed absorption bands at 3401, 3338, 2220, and 1653 cm^−1^, corresponding to hydroxyl, amino, cyano, and carbonyl groups, respectively as in Figure 5.

#### 3.2.2. Proton Nuclear Magnetic Resonance (HNMR) of Compound (ACR-3) 

1H NMR (DMSO-d6): δ 6.76–7.99 (m, 9H, Ar-H), 8.24 (s, 1H, =CH), 9.38 (s, 1H, OH), and 10.17 (s, 1H, N–H).

The ^1^H NMR showed multiple signals at *^TM^* 6.76–7.99 ppm for aromatic protons, in addition to singlet signals at *^TM^* 8.24, 9.38, and 10.17 ppm for the methine, OH, and NH protons, respectively as in Figure 6.

#### 3.2.3. Mass Spectroscopy (ACR-3)

MS *m*/*z* (%): 264 (M+, 40.25), 259 (36.22), 245 (62.84), 205 (58.23), 174 (51.07), 113 (56.05), 95 (74.61), 61 (100.00), and 56 (87.29). The MS exhibited a molecular ion peak at *m*/*z* = 264 (M^+^, 40.25%), congruous to a molecular formula C_16_H_12_N_2_O_2_ as in Figure 7.

#### 3.2.4. Mass Loss Test (ML) 

Of the main credible methods, weight loss tests are among the most important processes when assessing the inhibitor’s capacity to protect the metal against corrosion, and this is evident in Table 2 and Table 3, which show the results extracted as a result of immersing copper samples in a solution of nitric acid at a concentration of 1.0 M, and then adding various concentrations of the compounds ACR-2, and ACR-3 at temperatures of 25–45 °C. These compounds reduce corrosion rates with increasing concentrations, which are linked with the adsorption state on the metal surface, which causes a rise in the inhibition of the capability of these substances on the copper surface [29,30]. Figure 8 and Figure 9 show that raising the inhibitor concentration increased the protection efficiency at all investigated concentrations. As a result, it is clear that the protection performance was dosage-dependent. On the examined coupon surface, a greater number of inhibitor molecules were adsorbed when inhibitor doses were increased, resulting in an improved protection performance. Because the adsorbed inhibitor molecules control and/or restrict the reaction sites, the coupon surface is protected from the corrosive solution [31]. While the inhibitor contains considerable ion pairs pairs of electrons, such as ion pairs on oxygen and nitrogen atoms, and pi-electrons, which are coordinately linked with copper atoms on the coupon surface, the corrosion may be slowed.

#### 3.2.5. Adsorption Isotherm Analysis

The inhibition pathway can be determined through the isotherm of adsorption, which provides a detailed explanation of this pathway. It is also possible to see the inhibitor and the metal interaction through isotherm diagrams. To determine the most suitable diagrams, Langmuir, Freundlich, and Temkin models were tested [32,33]. The value of the correlation coefficient, which approached the right one, is illustrated in Figure 10 and Figure 11. ACR-2 and ACR-3 molecules adsorb in 1.0 M nitric acid solutions with correlation coefficients and slopes close to 1, indicating Langmuir adsorption isotherms. The *K_ads_* values could be determined from the intersections of the lines on the *C*/*θ*-axis and *K_ads_* was connected to the standard adsorption free energy ∆*G°_ads_* as follows:(6)$$\frac{C}{\theta }=\frac{1}{K_{ads}}+c$$

The value of the adsorption constant *K_ads_* was calculated, and through it, the free energy ∆*G°_ads_* values were computed using the equation:(7)$$K=\frac{1}{55.5}exp\left(\frac{-\Delta G_{ads}}{RT}\right)$$

Table 4 and Table 5 illustrate that the ACR-2 and ACR-3 molecules are thought to be electrostatically interconnected in an acidic solution when the Δ*G°_ads_* value is around −20 kJ mol^−1^ or lower, while a value of −40 kJ mol^−1^ or higher indicates a charge exchange or transfer between the copper surface and the ACR-2 and ACR-3 molecules [34]. From the values gotten from ∆*G°_ads_*, the presence of physical adsorption was confirmed. The enthalpy values resulting from electrostatic interactions between charged molecules and charged metal (physisorption) range up to 41.9 kJ mol^−1^, whereas those, resulting from chemisorption, range up to about 100 kJ mol^−1^. Physisorption produces molecules with small absolute enthalpy values. When the examined chemicals are present, ∆*S°_ads_* values are large and negative, indicating the increase in the ordering on copper surface [35,36].

#### 3.2.6. Potentiodynamic Polarization Test

By following the cathodic and anodic curves, adding different concentrations of inhibitors will display a declining trend in the current, which indicates the tight restriction of the anodic and cathodic interactions, as shown in Figure 12 and Figure 13. In addition, the lines are parallel when the inhibitors are added with the blanks, which proves that the reaction mechanism did not change by adding the inhibitors [37,38]. Additionally, there was a noticeable jump in the slope of the anode curve when changing the voltage values to the positive direction, which confirms the change in the absorbance values of the inhibitor. In addition, it can be observed that the polarization curve is small, which indicates the arrival of ACR-3 and ACR-2 to the equilibrium state on the surface. Table 6 shows the coefficients of the parameters from corrosion potential (*E_corr_*), anodic and cathodic slope (*βa*, *βc*), *i_corr_*, and *θ*; through it, a significant decrease in the values of *i_corr_* can be observed with the addition of different concentrations of ACR-2 and ACR-3, the increase of which leads to a reduction in the current density to the minimum [39]. In addition to increasing the values of the inhibition coefficient, which confirms the ability of the compounds ACR-2 and ACR-3 to provide a highly efficient inhibitory performance for copper in nitric acid owing to the adsorption of a layer of the inhibitor on the metal’s surface. Additionally, a small change in the values of *E_corr_*, which proves the mixed character of these compounds [40,41]. Inferia et al. report that the inhibitors are either cathodic or anodic types depending on the *E_corr_* value shift, whereas if *E_corr_* value shifts below 85 mV, the inhibitor is considered a mixed type. This study reveals that the studied inhibitor is a mixed-type inhibitor, as measured by a maximum displacement in *E_corr_* of 85 mV [42]. 

#### 3.2.7. Electrochemical Impedance Spectroscopy (EIS) Tests

The results of AC impedance were extracted, which are presented in Figure 14 and Figure 15 for the ACR-2 and ACR-3 compounds at 25 °C. As a result of the addition of these compounds, the rate of charge transfer was delayed, which is represented in the first incomplete circle, while the other half circle appears as a consequence of the heterogeneity of the copper surface [43]. Accordingly, the diameter of the halves of the circles gradually increases with raised doses of inhibitors owing to the dissolution of nitrate ions caused by ACR-2 and ACR-3. Calculating the capacitance of the double layers as follows [44]:(8)$$C_{dl}=Y_{0}{\left(2\pi f_{max}\right)}^{n-1}$$
where *Y*_0_ and *f_max_* are the magnitude of the CPE and frequency at which the imaginary component of the impedance reaches the maximum value, respectively. *n* is an adjustable factor, *n* = *α*/(*π*/2), and *α* is the phase angle. As the concentrations of the compounds increase, the values of *C_dl_* decrease because the thickness of the adsorbed layer on the copper surface increases [45]. Based on the impedance data in Table 7 and Table 8, it can be concluded that inhibitor doses and hence inhibitory efficacy in acidic solutions increase with increasing *R_ct_* values. The creation of a protective coating at the metal solution contact can be attributed to this. There was only one charge transfer mechanism during copper dissolution, determined by the appearance of a semicircle without any impact on the molecule being studied. A decrease in double-layer capacitance was observed with increasing inhibitor doses [46]. This can be attributed to the density of the electrical double layer. 

#### 3.2.8. Molecular Modeling

Quantum chemical simulations utilizing the DFT mode were carried out on acrylamide derivatives and the components’ molecular function under investigation. DFT was used to obtain the optimized molecular geometries of the corrosion inhibitor to test the effect the molecular structure has on its performance [47]. Molecular models were created using Gauss View 2009 and DFT calculations were performed using a basis set 6-31G (d). Quantum chemical parameters are extensively described elsewhere, including a detailed calculation method. Understanding how an inhibitor interacts with a metallic surface depends on the electron-donating properties of molecules and surfaces. In Figure 16, you can see both the lowest unoccupied molecular orbital surfaces (LUMO) and the highest occupied molecular orbital surfaces (HOMO). HOMO surfaces indicate which sites of the molecule are most likely to donate electrons to the metal, whereas LUMO surfaces indicate that electrons from the metal are likely to be regifted to the inhibitor [48,49]. The HOMO in ACR-2 and ACR-3 is mainly centered around nitrogen, oxygen, and the π bonding on the benzene ring where resonance occurs. Meanwhile, the LUMO is concentrated around the carbon atoms. The ΔE value can be used to evaluate the inhibitor molecule’s reactivity to the metal atom; lower ΔE values may imply improved inhibition efficiency [50,51]. From these DFT calculations and the trends described, it was determined that ACR-2 (ΔE = 3.082) behaved as a superior corrosion inhibitor relative to ACR-3 (ΔE = 3.510) due to its lower ΔE. On the other hand, the dipole moment (DM) does not play an essential role in determining the efficiency of inhibition [52]. As well as this, Mulliken atomic charges may also provide information regarding the actual sites where inhibitor molecules adsorb on the surface of the cell. The charges of chelating atoms have been investigated in a variety of studies in an attempt to determine whether they influence the inhibitors’ ability to adhere to metallic surfaces. The higher the negative Mulliken charge of the adsorbed site, the better the possibility of an electron being donated to the partially filled or vacant d orbital of the metal. Mulliken charges summed into carbon and heavy atoms, i.e., N and O atoms for ACR-2 and ACR-3, are depicted in Figure 16. According to ACR-2, the highest Mulliken Atomic charges are observed at 8N, 7O, 19O, and 20N. Therefore, it is reasonable to assume that the charge-deficient metallic atom will be attracted to the appropriate atom within the anionic moiety of the compounds studied.

#### 3.2.9. MD Simulation

The position at which ACR-3 and ACR-2 were adsorbed onto the Cu (1 1 1) contact is shown in Figure 17. As a result of the parallel adsorption of ACR-3 and ACR-2, the copper substrate is highly protected [53]. When comparing the data in Table 9, it is evident that compound ACR-2 has higher adsorption energy than compound ACR-3, indicating that compound ACR-2 is more effective at shielding the copper surface than compound ACR-3. According to Table 9, adsorption energies, rigid absorption energies, and strain energies were determined from the MD simulation. Adsorption energy (*E_ads_*) was calculated which indicates how strongly the inhibitor binds to its substrate surface (copper). Inhibitory molecules are said to release rigid adsorption energy when they adsorb to metal surfaces, whereas on the other hand, deformation energy is released when the adsorbed component relaxes at the surface [54]. ACR-2 and ACR-3, *E_ads_* negative indicate that strong and spontaneous adsorption occurred on the copper substrate. The results of Table 9 indicate that ACR-2 has greater adsorption energy than ACR-3, indicating it will perform better as an inhibitor for the protection of copper in 1.0 M nitric acid than ACR-3. This theoretical analysis supports previous studies and is accurate for all available data. Lastly, ACR-2 and ACR-3 inhibitors offer great protection against corrosion.

#### 3.2.10. Scanning Electron Microscope (SEM) Analysis

Figure 18 illustrates the sample of Cu coupons exposed for 5 hours to 1.0 M nitric acid in the lack (without an inhibitor) and the existence of an inhibitor (20 × 10^−5^ M). Figure 18a of the SEM photos shows how the corrosion product caused severe damage to the Cu surface. Figure 18b,c show how the Cu surface improved by adding inhibitors to its maximum concentration [55].

#### 3.2.11. Mechanism of Inhibition 

An experimental study and theoretical calculation can be used to demonstrate the adsorption of Cu by calculating the type of functional moieties, electron density, and the charge on the Cu surface. It has been reported that the Cu surface is positively charged in the nitric acid solution, i.e., several positive charges are present on the Cu surface [56]. The positively charged Cu surface prefers NO_3_− adsorption to create a negatively charged surface, making the adsorption of the cations in solution easier [57]. Organic acrylamide compounds may be protonated in solution since the N and O electron pairs are unshared. A phenomenon known as physisorption occurs when protonated molecules bond electrostatically to specimen surfaces. While waiting for more adsorption, these inhibitor compounds can be covalently linked (chemisorption). ML and electrochemical analysis revealed the following percentage η% of the two acrylamide derivatives studied: ACR-2 > ACR-3 is the most effective inhibitor due to its larger molecular size and the presence of OCH_3_ groups, which act as electron donors.

## 4. Conclusions

Experimental experiments proved the effectiveness of 2-acrylamide derivatives in inhibiting copper corrosion in 1.0 M HNO_3_ solution as mixed-type inhibitors. By adsorbing these compounds onto copper surfaces, these compounds exhibited Langmuir adsorption isotherms, which corresponded to the inhibition process. The spontaneity of the reaction was inferred due to the presence of the negative value of Δ*G*°*_ads_*. DFT revealed an increase in the inhibition strength of the compounds ACR-3 and pc ACR-2 due to the ability to donate and gain electrons from the highest occupied energy (HUMO) orbital to the lowest unoccupied orbital (LUMO). The corrosion process depends on the relative ability of the compounds to donate electrons to the metal. In the end, the results were found to agree with the measurements. 

## Figures and Tables

**Figure 1 nanomaterials-12-03685-f001:**
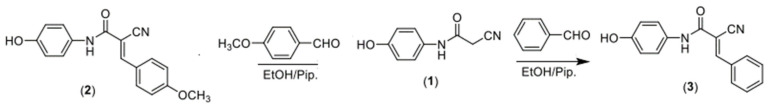
Synthesis of ACR-2 and ACR-3 compounds.

**Figure 2 nanomaterials-12-03685-f002:**
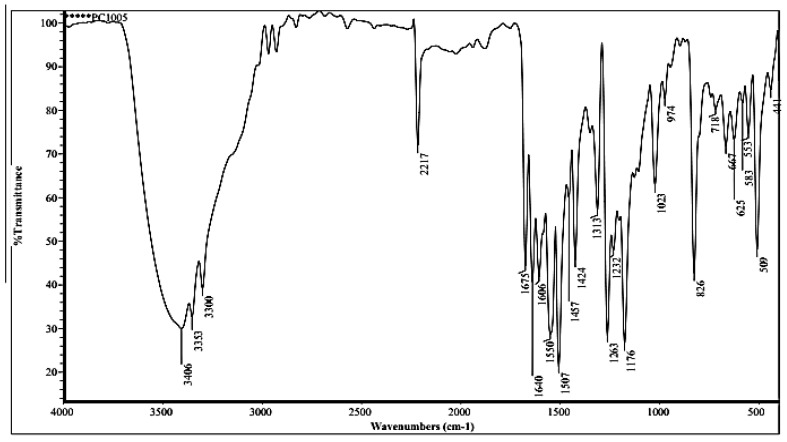
FTIR of ACR-2.

**Figure 3 nanomaterials-12-03685-f003:**
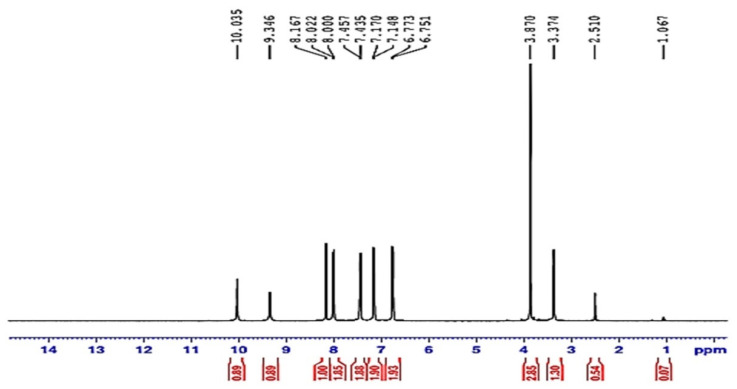
1H-NMR spectrum of ACR-2.

**Figure 4 nanomaterials-12-03685-f004:**
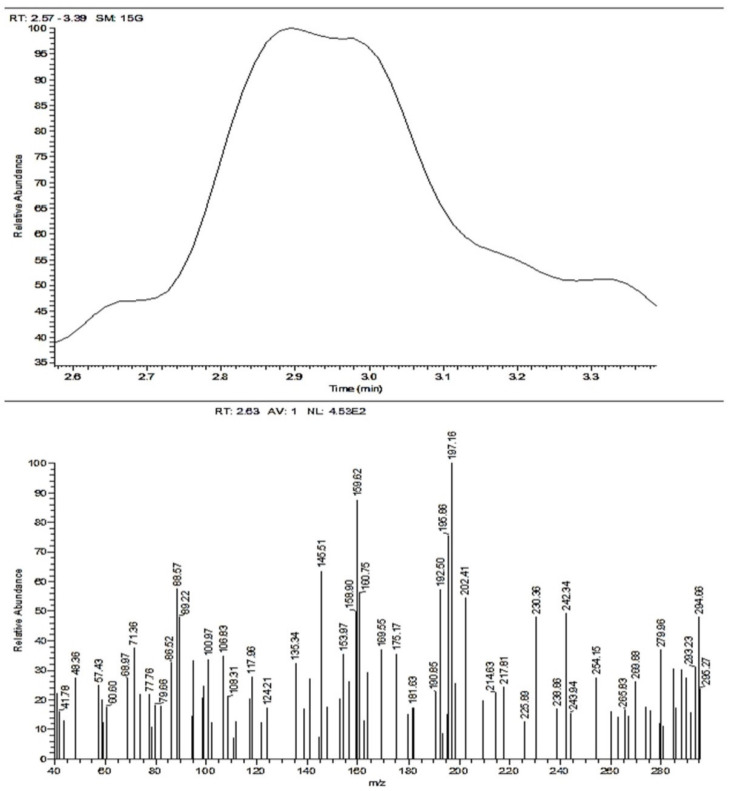
Mass spectroscopy of ACR-2.

**Figure 5 nanomaterials-12-03685-f005:**
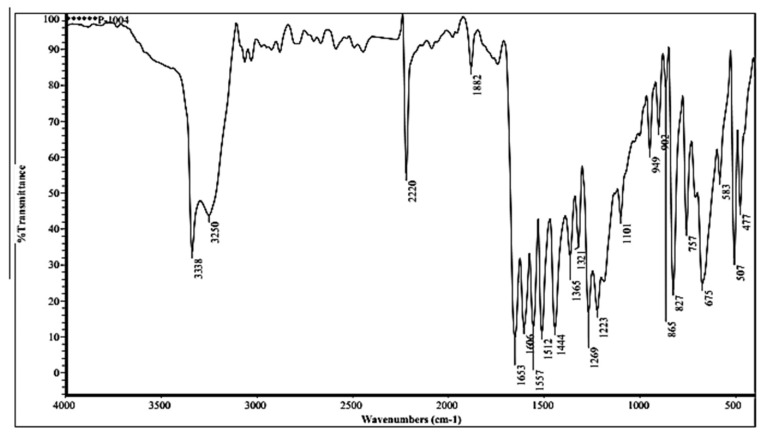
FTIR of ACR-3.

**Figure 6 nanomaterials-12-03685-f006:**
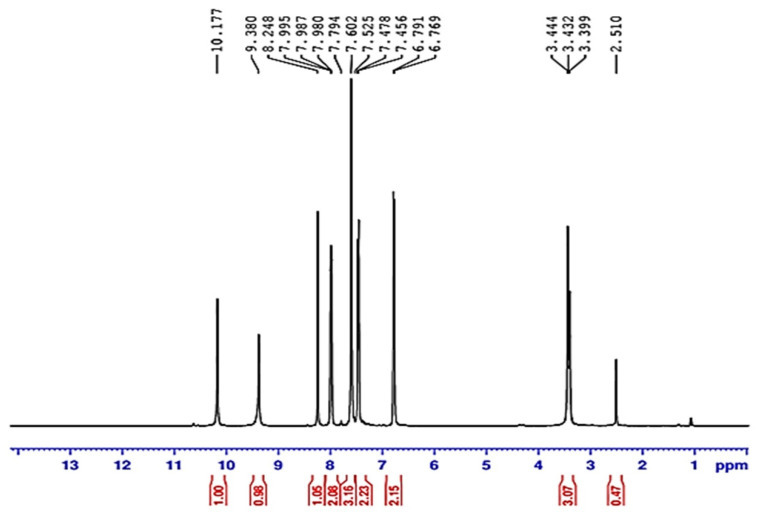
The 1H-NMR spectrum of ACR-3.

**Figure 7 nanomaterials-12-03685-f007:**
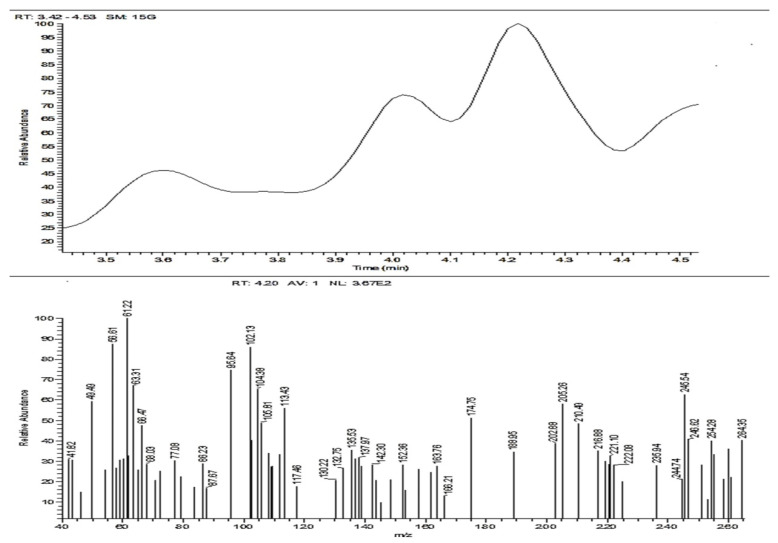
Mass Spectroscopy of ACR-3.

**Figure 8 nanomaterials-12-03685-f008:**
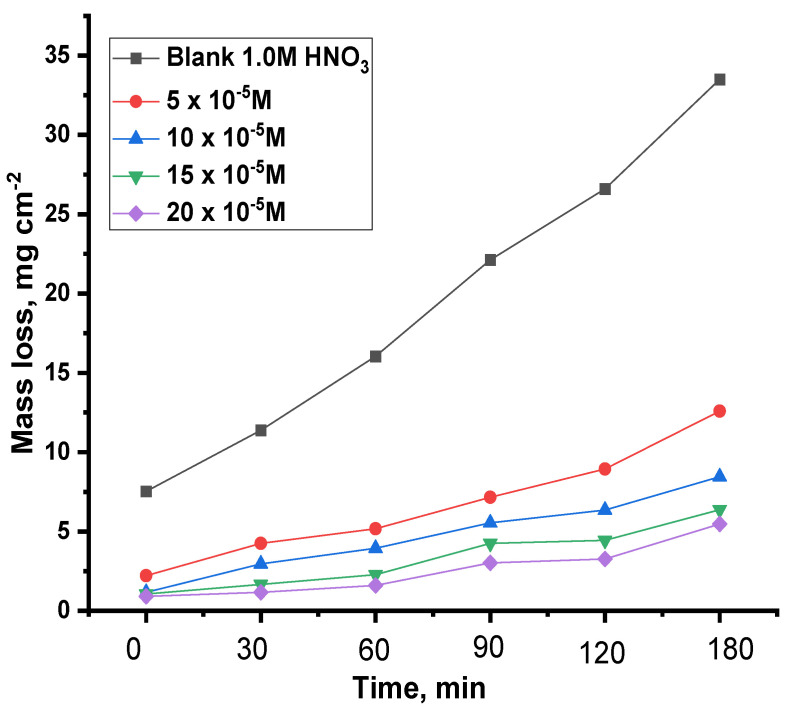
Cu mass loss under various concentrations of ACR-2 in 1.0 M HNO_3_ with and without immersion times.

**Figure 9 nanomaterials-12-03685-f009:**
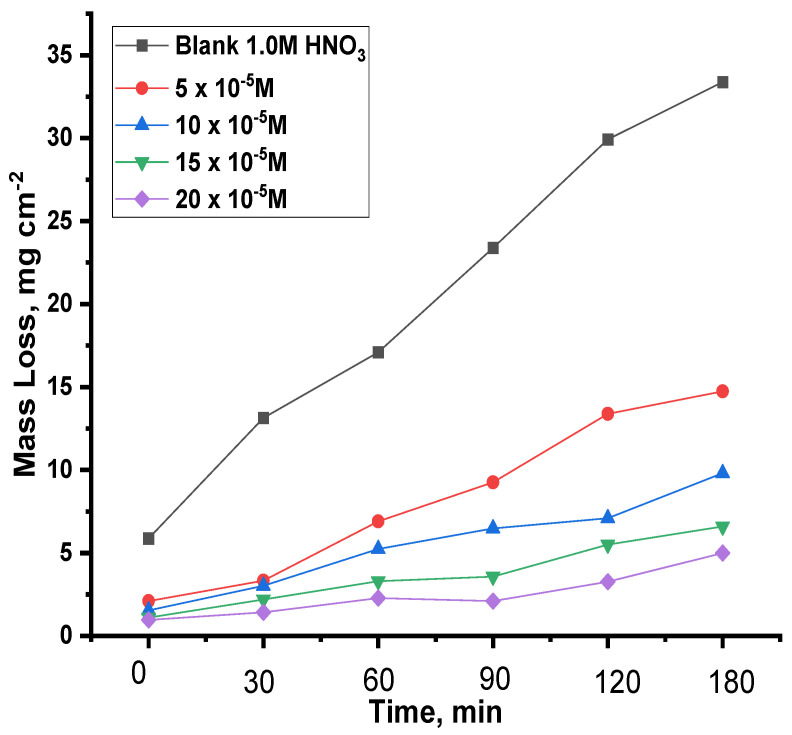
ACR-3 at 25 °C and various immersion times in 1.0 M HNO_3_ with and without ACR-3.

**Figure 10 nanomaterials-12-03685-f010:**
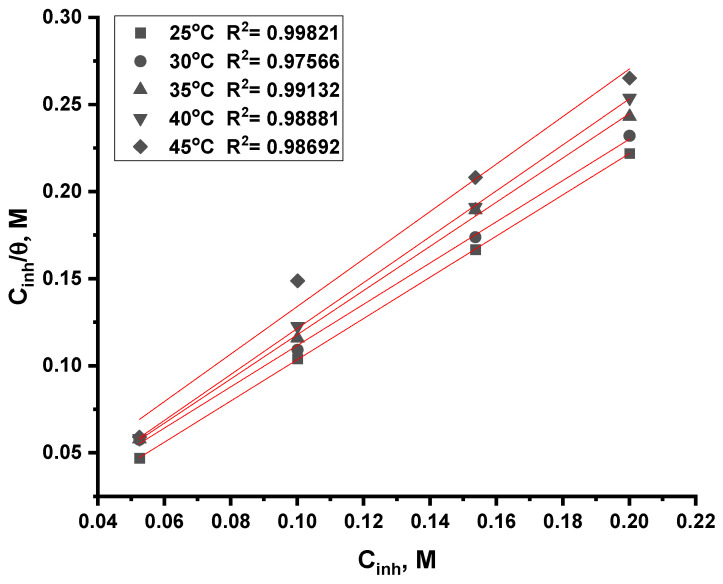
Langmuir adsorption plot as *C*/*θ* vs. C, M of ACR-2 for corrosion of copper in 1.0 M HNO_3_ solution at all temperatures.

**Figure 11 nanomaterials-12-03685-f011:**
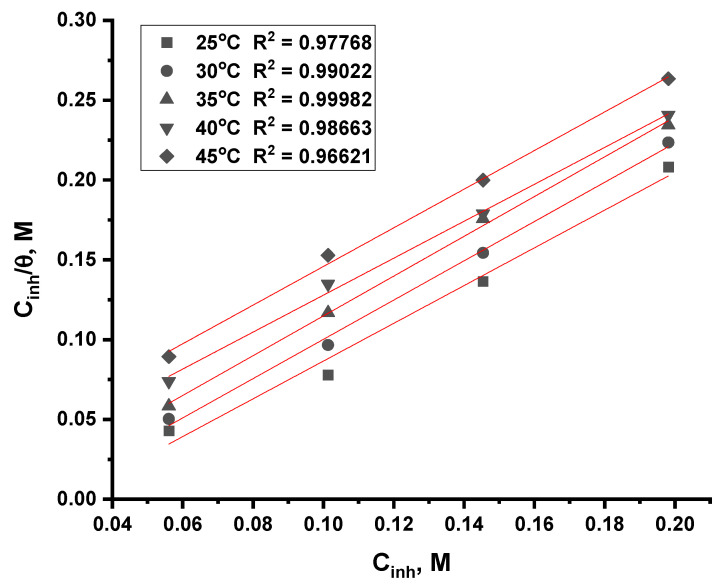
Langmuir adsorption plot as *C*/*θ* vs. *C*, *M* of ACR-3 for corrosion of copper in 1.0 M HNO_3_ solution at all temperatures.

**Figure 12 nanomaterials-12-03685-f012:**
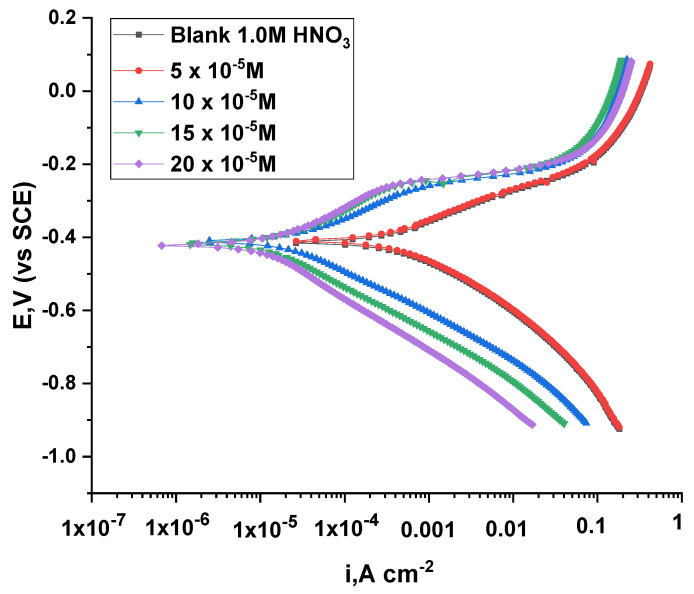
Plots of PP of Cu dissolution with and without different doses of ACR-2.

**Figure 13 nanomaterials-12-03685-f013:**
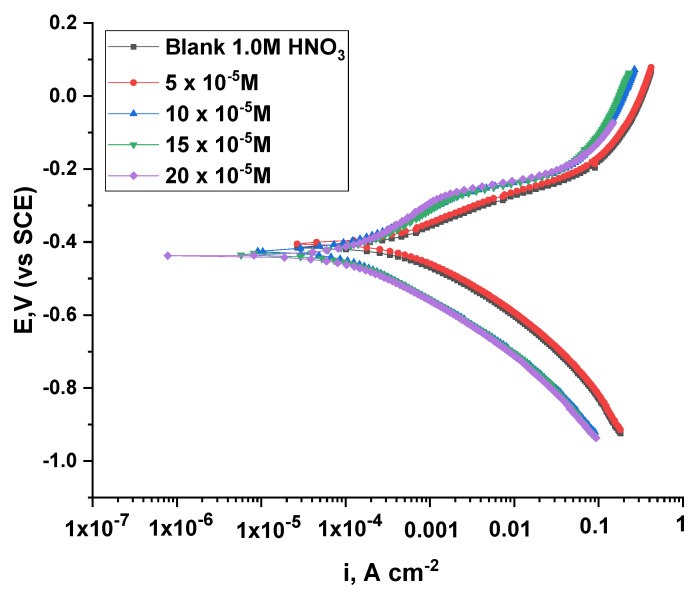
Plots of PP of Cu dissolution with and without different doses of ACR-3.

**Figure 14 nanomaterials-12-03685-f014:**
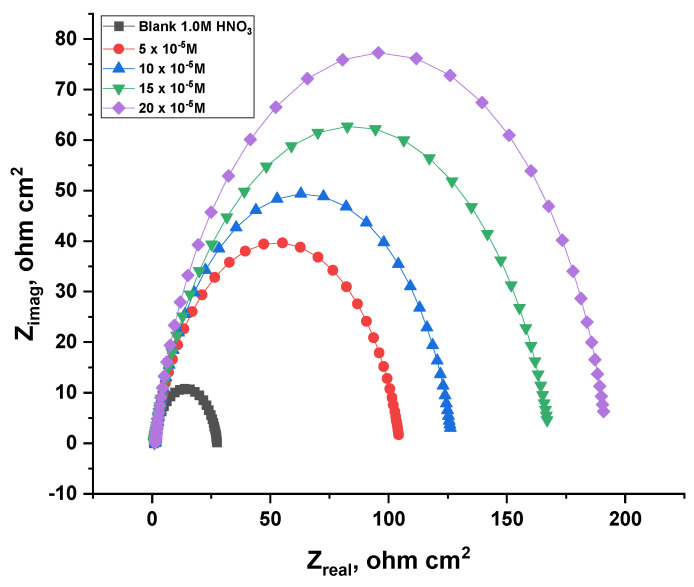
An analysis of the Nyquist plot for copper dissolution in 1.0 M HNO_3_ solution in the absence and presence of numerous doses of ACR-2.

**Figure 15 nanomaterials-12-03685-f015:**
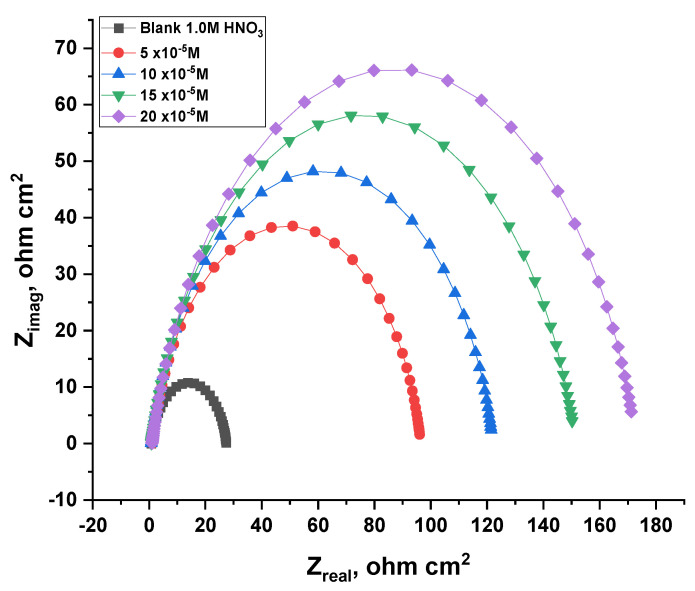
An analysis of the Nyquist plot for copper dissolution in 1.0 M HNO_3_ with and without different doses of ACR-3.

**Figure 16 nanomaterials-12-03685-f016:**
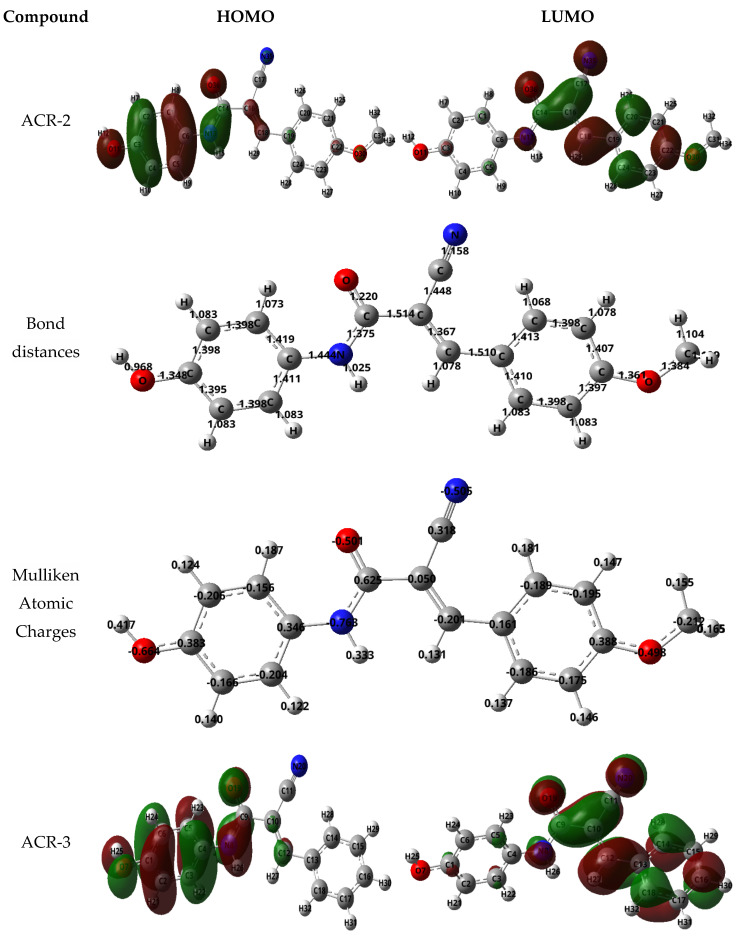
Bond distances, Mulliken atomic charges, HOMOs, and LUMOs, of compounds ACR-2 and ACR-3.

**Figure 17 nanomaterials-12-03685-f017:**
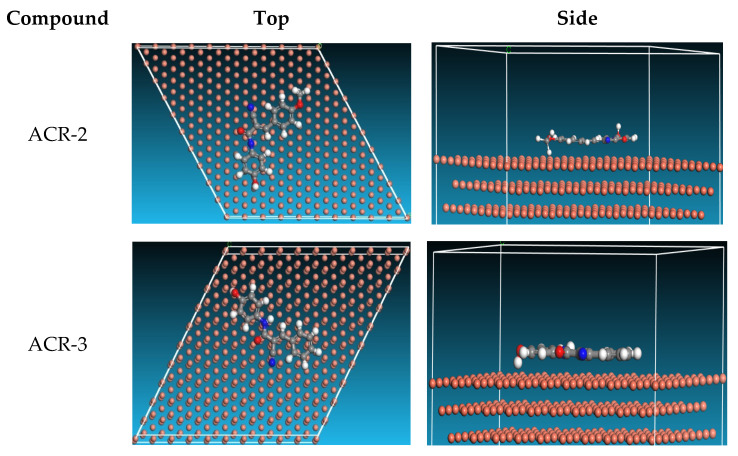
Adsorption of acrylamide compounds on Cu surface.

**Figure 18 nanomaterials-12-03685-f018:**
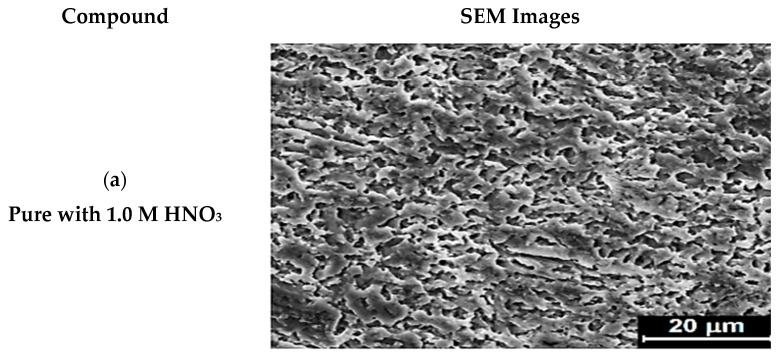
SEM images from Cu (**a**) after soaking time for 5 h in 1.0 M HNO_3_, and (**b**) in the 20 × 10^−5^ M inhibitor ACR-2 (**c**) in the 20 × 10^−5^ M inhibitor ACR-3.

**Table 1 nanomaterials-12-03685-t001:** Chemical composition of the brass alloy in weight %.

Element	Sn	Fe	Ni	Pb	As	Cu
Weight %	1.44	0.89	0.27	2.5	37.5	56.8

**Table 2 nanomaterials-12-03685-t002:** Different measurements of inhibition for the corrosive effects of 1.0 M HNO_3_ on copper with different concentrations both with and without ACR-2 at 25 °C.

ACR-2 × 10^−5^ M	5	10	15	20
*θ*	0.7694	0.7898	0.8079	0.8350
η%	76.94	78.98	80.79	83.50

**Table 3 nanomaterials-12-03685-t003:** Different measurements of inhibition for the corrosive effects of 1.0 M HNO_3_ on copper with different concentrations both with and without ACR-3 at 25 °C.

ACR-3 × 10^−5^ M	5	10	15	20
*θ*	0.271	0.448	0.626	0.72
η%	27.1	44.8	62.6	72.0

**Table 4 nanomaterials-12-03685-t004:** Thermodynamic adsorption coefficients of ACR-2 on Cu substrate in 1.0 M HNO_3_.

Temp., °C	*K_ads_*	log*k_ads_*	ΔG	ΔH	ΔS
25	−1.15066	0.0609	−10.3004	−0.008848	0.034595
30	−1.17425	0.0698	−10.5244		0.034734
35	−1.16331	0.0657	−10.674		0.034656
40	−1.1763	0.0705	−10.8762		0.034748
45	−1.45069	0.1615	−11.604		0.03649

**Table 5 nanomaterials-12-03685-t005:** Parameters of thermodynamic adsorption of ACR-3 on Cu surface in 1.0 M HNO_3_.

Temp., °C	*K_ads_*	log*k_ads_*	ΔG	ΔH	ΔS
25	−1315.1	3.1189	−27.7486	−0.13403	0.092666
30	−232.558	2.3665	−23.8491		0.07871
35	−1996.57	3.3002	−29.749		0.096588
40	−925.926	2.9665	−28.232		0.090198
45	−34.4708	1.5374	−19.9815		0.062835

**Table 6 nanomaterials-12-03685-t006:** Corrosion parameters obtained from potentiodynamic polarization of Cu in 1.0 M HNO_3_ containing numerous doses of inhibitors.

Comp.	Conc. × 10^−5^ M	*i_corr_*_,_ μA cm^−2^	−*E_corr_*, mV vs. SCE	*β_a_* mV dec^−1^	*β_c_* mV dec^−1^	C.R mpy	*θ*	η%
Blank	0.0	494.0	413	123	181	131	-	-
ACR-2	5	108.0	423	116	180	28	0.781	78.1
	10	97.0	421	123	191	20	0.804	80.4
	15	71	428	122	186	13	0.856	85.6
	20	57.0	430	127	179	8	0.885	88.5
ACR-3	5	115.0	423	125	190	47	0.767	76.7
	10	78	421	119	183	31	0.842	84.2
	15	62.0	423	120	189	28	0.874	87.4
	20	55.0	418	121	181	23	0.889	88.9

**Table 7 nanomaterials-12-03685-t007:** Data for the corrosion of Cu in 1.0 M HNO_3_ solution in in the absence and presence of numerous doses of ACR-2 from the EIS.

Inhibitor A, M	*C_dl_*, µF cm^−2^	*R_p_*, Ω cm^2^	*θ*	η%
Blank	165	27	-	-
5	155	104	0.744	74.4
10	144	126	0.788	78.8
15	125	168	0.841	84.1
20	106	192	0.861	86.1

**Table 8 nanomaterials-12-03685-t008:** Statistics from the EIS on the corrosion of Cu in 1.0 M HNO_3_ in the absence and presence of numerous doses of ACR-3.

Inhibitor B, M	*C_dl_*, µF cm^−2^	R_ct,_ Ω cm^2^	*θ*	η%
Blank	165	27	-	-
5	95	150	0.719	71.9
10	115	146	0.768	76.8
15	150	140	0.822	82.2
20	172	135	0.845	84.5

**Table 9 nanomaterials-12-03685-t009:** Monte Carlo simulation parameters of adsorption of acrylamide compound on Cu (1 1 1) surface.

Structure	Total Energy	Adsorption Energy	Rigid Adsorption Energy	Deformation Energy	*E_ads_*: Compound
Cu (1 1 1) −1(ACR-2)	−154.37	−94.72	−96.49	1.770	−94.72
Cu (1 1 1) −1(ACR-3)	−164.41	−83.73	−87.94	4.215	−83.73

## Data Availability

The authors confirm that the data supporting this study are available.

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
