# Peer review of "A Study of the Synthesis and Characterization of New Acrylamide Derivatives for Use as Corrosion Inhibitors in Nitric Acid Solutions of Copper"

_nanomaterials, 2022, doi:10.3390/nano12203685_

Round 1

Reviewer 1 Report

This study investigated two acrylamide derivatives as corroison inhibitors. The abstract needs to mention the names of the two compounds. Figure 2 to 7 and figure 10 to 11 even do not have any explainations. The other figures lack detailed and insightful discussion. This paper is recommended to be rejected.

Author Response

Supalak Manotham
Assistant Editor

Nanomaterials

9 Oct 2022

Subject: Revision and resubmission of manuscript nanomaterials-1955868

Dear Editor and reviewers

Thank you for your letter and the opportunity to revise our manuscript entitled” " A study of the synthesis and characterization of new acrylamide derivatives for use as corrosion inhibitors in nitric acid solutions of copper" (Manuscript Number: nanomaterials-1955868). The suggestions offered by the reviewers have been immensely helpful. We have studied the comments carefully and have made corrections marked in red in the revised manuscript. The main corrections and adjustments in the paper and the response to the reviewers’ comments are as follows.

Reviewer 1:

  • The abstract needs to mention the names of the two compounds.

Response: Thank you. The names of the two compounds and their abbreviations have been added to the abstract.

  • Figures 2 to 7 and figures 10 to 11 even do not have any explanations and the other figures lack detailed and insightful discussion.

Response: Thank you. More explanations have been added to all figures in the manuscript.

Reviewer 2 Report

The work “A study of the synthesis and characterization of new acrylamide derivatives for use as corrosion inhibitors in nitric acid solutions of copper” by Ahmed Abu-Rayyan, Huda S. AlSalem, Zainab A. Jabarah, , Nahlah A. Alhadhrami, Amira K. Hajri, Abeer Abdulaziz H. Bukhari, Mohamed. M. Waly and A.M. Salem is a combined experimental and computational study (to complement) on acrylamides to be used as corrosion inhibitors.

The first impression is bad just looking at the list of names above because it is easy to see multiple errors: spaces, missing commas… and throughout the manuscript from the Abstract those typos continue: not using subscripts for example (HNO3). In the text the same happens, and expressions such as “kilojoules/mol” instead of kJ/mol are strange. Hundreds of wrong words like “orbit”.

To try to generate interest to the reader and that acrylamides are not only used to prevent corrosion in particular solutions, an attempt should be made to broaden the horizon of the same, and to talk about corrosion in a broader sense, related to metals and polymers (Corros. Sci. 2011, 53(5), pp. 1680-1689, DOI: 10.1016/j.corsci.2011.01.019; Polymers 2022 , 14, 2856, https://doi.org/10.3390/polym14142856; Appl. Surf. Sci. 2015, 353, 173-183, DOI: 10.1016/j.apsusc.2015.06.128; Int. J. Mol. Sci. 2021, 22, 13383, DOI: 10.3390/ijms222413383; Appl. Surf. Sci. 2018, 441, 895-913, DOI: 10.1016/j.apsusc.2018.02.012). Again the format of references is a chaos, with journals abbreviated, other ones not (most of them), then dots that should not be present…

The Figures could be more convincing, for example Figure 3 including the distribution of peaks related to each chemical moiety, with arrows for example (same for Figure 6). And the same for Figure 5, corresponding to the FTIR spectroscopy.

The experimental characterization is not great, but it is descriptive, and if Figures were improved or even gathered, this part is OK. The problem are the calculations, Figure 16 is useless and should be moved to the SI, but anyway it is quite useless, since you cannot put an optimized structure without any distance or angle, with those blue brackgrounds, with labels on atoms that you would need a microscope to see them. And the figures of the frontier molecular orbitals are descriptive, but which is the interest? How is translated if one or another region HOMO and LUMO are referred to? The dipole moment is interesting from this part of DFT calculations. DeltaE means the HOMO-LUMO gap? This should be corrected and use 3 decimals for both values. In the conclusions remove the definition of HOMO and LUMO that were yet explained before. But explain what it means to “donated and gain electrons from the HUMO (I suppose HOMO) to the LUMO, or correct the sentence.

Overall, the format is really “to be improved”, figures must be gathered (for example 14 and 15), text has to be checked again.

Where are the computational details? Which is the level of theory?

Since there is a long list of flaws throughout the manuscript, if the paper was considered for publication, it would require a deep revision first.

Author Response

Supalak Manotham
Assistant Editor

Nanomaterials

9 Oct 2022

Subject: Revision and resubmission of manuscript nanomaterials-1955868

Dear Editor and reviewers

Thank you for your letter and the opportunity to revise our manuscript entitled” " A study of the synthesis and characterization of new acrylamide derivatives for use as corrosion inhibitors in nitric acid solutions of copper" (Manuscript Number: nanomaterials-1955868). The suggestions offered by the reviewers have been immensely helpful. We have studied the comments carefully and have made corrections marked in red in the revised manuscript. The main corrections and adjustments in the paper and the response to the reviewers’ comments are as follows.

Reviewer 2:

  • The first impression is bad just by looking at the list of names above because it is easy to see multiple errors: spaces, missing commas… and throughout the manuscript from the Abstract those typos continue: not using subscripts for example (HNO3). In the text, the same happens, and expressions such as “kilojoules/mol” instead of kJ/mol are strange. Hundreds of wrong words like “orbit”.

Response: Thank you.

All symbols have been modified and use of subscripts and expressions containing errors.

  • To try to generate interest in the reader and that acrylamides are not only used to prevent corrosion in particular solutions, but an attempt should also be made to broaden the horizon of the same, and to talk about the corrosion in a broader sense, related to metals and polymers (Corros. 2011, 53(5), pp. 1680-1689, DOI: 10.1016/j.corsci.2011.01.019; Polymers 2022 , 14, 2856, https://doi.org/10.3390/polym14142856; Appl. Surf. Sci. 2015, 353, 173-183, DOI: 10.1016/j.apsusc.2015.06.128; Int. J. Mol. Sci. 2021, 22, 13383, DOI: 10.3390/ijms222413383; Appl. Surf. Sci. 2018, 441, 895-913, DOI: 10.1016/j.apsusc.2018.02.012). Again the format of references is a chaos, with journals abbreviated, other ones not (most of them), then dots that should not be present…

Response: Thank you.

It described more about the inhibitors that are used to protect metals from corrosion and more about the polymers used in the inhibition processes and added this part in the introduction to the research and added with references in the manuscript.

  • The experimental characterization is not great, but it is descriptive, and if Figures were improved or even gathered, this part is OK. The problem are the calculations, Figure 16 is useless and should be moved to the SI, but anyway it is quite useless, since you cannot put an optimized structure without any distance or angle, with those blue brackgrounds, with labels on atoms that you would need a microscope to see them. And the figures of the frontier molecular orbitals are descriptive, but which is the interest? How is translated if one or another region HOMO and LUMO are referred to? The dipole moment is interesting from this part of DFT calculations. DeltaE means the HOMO-LUMO gap? This should be corrected and use 3 decimals for both values. In the conclusions remove the definition of HOMO and LUMO that were yet explained before. But explain what it means to “donated and gain electrons from the HUMO (I suppose HOMO) to the LUMO, or correct the sentence.

Response: Thank you.

All Figures in the experimental characterization have been improved and added to, where the numbers and type of atoms have been added, the spaces between the atoms have been added to the improved structure, the addition of a Mulliken  Atomic charges, and the HOMO, LUMO, Dipole moment and the ΔE (energy gap) between HOMO and LUMO have also been clarified more and more comprehensively, and decimals have been modified and three decimals are used for all values.

Round 2

Reviewer 1 Report

Why did you conduct this research? You cannot say it is a hot topic, so you conduct this research. Please revise introduction.

Author Response

Reviewer 1:

1-         Why did you conduct this research? You cannot say it is a hot topic, so you conduct this research. Please revise the introduction.

Response: Thank you.

The introduction has been revised and the reason for conducting this research has been added.

Reviewer 2 Report

The first revision of the manuscript has improved the manuscript, but this was easy because the first was really weak in format terms.

Still there are “flaws” in format, for example in the new piece of text, line 249: “100 kJ mol-135.” And in the same text, line 242 it is not clear what it means “Table 4,5” and the same in line 308 “in Tables 7,8,” Probably it would be better Tables 4 and 5 and then Tables 7 and 8.

The computational part in DFT is extremely weak, and maybe in the first report this could seem that only in terms of format, but also in terms of chemistry. It is so weak that it is based on charges that actually then are useful to describe the potential role of heteroatoms, but I ask to remove Table 9 and simply state that the HOMO LUMO gap is larger for ACR-3 and any other information the authors consider important. And for Figure 16 that can be moved to the SI, but since there is no SI, it should be improved. Actually, instead of “Optimized structure” it should be stated “Bond distances” because also the next figure with charges is from an optimized structure. And then give detail of how Mullikan and Milliken charges are calculated. I only know Mulliken charges.

Since I did not reject the paper in the first round, I give the authors a second chance to improve the paper even more.

Author Response

Reviewer 2:

1-         The first revision of the manuscript has improved the manuscript, but this was easy because the first was really weak in format terms.

Still there are “flaws” in format, for example in the new piece of text, line 249: “100 kJ mol-135.” And in the same text, line 242 it is not clear what it means “Table 4,5” and the same in line 308 “in Tables 7,8,” Probably it would be better Tables 4 and 5 and then Tables 7 and 8.

Response: Thank you.

Format 'flaws' have been reviewed and corrected.

2-         The computational part in DFT is extremely weak, and maybe in the first report this could seem that only in terms of format, but also in terms of chemistry. It is so weak that it is based on charges that actually then are useful to describe the potential role of heteroatoms, but I ask to remove Table 9 and simply state that the HOMO LUMO gap is larger for ACR-3 and any other information the authors consider important. And for Figure 16 that can be moved to the SI, but since there is no SI, it should be improved. Actually, instead of “Optimized structure” it should be stated “Bond distances” because also the next figure with charges is from an optimized structure. And then give detail of how Mullikan and Milliken charges are calculated. I only know Mulliken charges.

Response: Thank you.

1-         Table 9 is removed and simply talks about the HOMO LUMO gap with some simple information added to improve this theoretical part.

2-         There is no SI, but the graphics have been improved upon your request.

3-         " Bond distances " has been replaced by " Optimized structure "

4-         sorry, by mistake the " Mullikan and Milliken charges " was inadvertently added .so, I removed it.

3-         Since I did not reject the paper in the first round, I give the authors a second chance to improve the paper even more.

 Response

Thank you very much for giving me a second chance and working on strengthening the manuscript after correcting those mistakes.